# Antibody signatures elicited by potent and subpotent whole-cell pertussis vaccines in mice

Yetunde Adewunmi,[1] Jennifer Doering,[1] Prashant Kumar,[2] Jozelyn V. Pablo,[3] Andy A. Teng,[3] Vu Huynh,[3] Kathryn Secrist,[2] David B. Volkin,[2] Sangeeta B. Joshi,[2] Joseph J. Campo,[3] Nicholas J. Mantis[1]

**ABSTRACT**    Inactivated, whole-cell pertussis (wP) vaccines remain at the frontline in the global fight against the resurgence of whooping cough, especially in low- and middle-income countries. However, the reliance on the intracerebral mouse potency test (ic-MPT or Kendrick assay) as the standard batch release assay is extremely burdensome for commercial wP vaccine production. The ic-MPT is technically challenging, labor intensive, and incongruous with modern animal welfare guidelines. Replacing the ic-MPT with a whole-cell *Bordetella pertussis* enzyme-linked immunosorbent assay, the so-called pertussis serology potency test, has shown promise but has been difficult to implement in practice. In this report, we tested the hypothesis that potent and subpotent wP vaccines have distinct serological profiles in mice that could be developed as a substitute for the ic-MPT. We established an accelerated decay (thermal stress) protocol in which wP, in the context of diphtheria-tetanus-whole-cell pertussis, was rendered >10-fold less effective than unstressed vaccine when evaluated in a mouse model of *B. pertussis* lung clearance following intranasal challenge. We then screened immune sera on a limited *B. pertussis* Tahoma I proteome array and identified >30 antigens whose antibody reactivity profiles either increased, decreased, or were unchanged as a function of wP potency. Moreover, virtually all the "indicator" antigens identified are known virulence factors or reactive with human convalescent sera, thereby establishing a potential link between wP potency and pertussis infection and immunity. These results support the development of a limited *B. pertussis* antigen array as a stability-indicating surrogate potency assay for the ic-MPT.

**IMPORTANCE** Whooping cough (pertussis) is a highly contagious respiratory disease caused by the Gram-negative bacterium, *Bordetella pertussis*. Globally, tens of millions of whole-cell pertussis (wP) vaccines are administered annually. Whole-cell pertussis vaccines are logistically complex to manufacture and get to market because of the need for each batch of vaccine to be evaluated in a highly laborious and challenging potency test known as the Kendrick assay, which involves mouse intracerebral challenges with *B. pertussis*. In this report, we describe efforts to develop a serology-based substitute for the Kendrick assay that relies on profiling antibody responses to wP vaccines.

**KEYWORDS**    vaccine, potency, proteome, mouse, pertussis

Pertussis (whooping cough) is a highly contagious respiratory infection caused by the Gram-negative bacterium, *Bordetella pertussis* (1–3). Pertussis is endemic in developed and low- and middle-income countries (LMICs), with epidemics occurring every 2–5 years despite high vaccination rates (4, 5). The estimated global burden of pertussis is 24.1 million cases and >160,000 deaths in children younger than the age of 5 (6). Although pertussis is widely considered a childhood illness due to its high disease severity and fatality in infants, clinically significant cases occur frequently

Address correspondence to Nicholas J. Mantis, Nicholas.mantis@health.ny.gov.

Yetunde Adewunmi and Jennifer Doering contributed equally to this article. The author order was determined alphabetically.

The authors declare no conflict of interest.

See the funding table on p. 17.

in adolescents and adults (7). Transmission occurs via airborne respiratory droplets generated by coughing or sneezing, and the resulting disease is characterized by uncontrollable violent coughing with a characteristic whoop for up to 4–12 weeks. Simultaneous immunization with pertussis antigen(s) combined with diphtheria (D) and tetanus toxoids (TT) remains the most effective way to reduce the disease burden of pertussis and therefore mortality in humans. Due to the reactogenicity associated with the use of whole-cell pertussis (wP) vaccines, the acellular pertussis (aP) vaccine has replaced the use of diphtheria-tetanus-whole-cell pertussis (DTwP) in many countries. Nevertheless, DTwP remains in widespread use globally for reasons related to cost, efficacy, and durability (2). Given the important role that DTwP vaccine has played in protecting the global population against pertussis, tetanus, and diphtheria, there are strong incentives to further increase the number of antigens in pediatric trivalent and pentavalent pertussis combination vaccines (i.e., DTwP-Hib-HepB) to improve global immunization coverage, affordability, and compliance (8).

However, the continued reliance on the intracerebral mouse potency test (ic-MPT or Kendrick assay) as the batch-release potency assay for wP vaccines is a challenge for getting wP-containing vaccines released for use in the market. The ic-MPT, commonly known as the Kendrick assay, is technically challenging (e.g., time consuming, labor intensive, and highly variable) and of questionable relevance to vaccine-induced immunity in humans (9–12). Moreover, the test is incompatible with current animal welfare guidelines because death is used as the endpoint. Considering the number of wP vaccines administered yearly and the corresponding number of animals required for the ic-MPT, there is a strong incentive to identify alternatives to replace or reduce the use of this assay for vaccine quality control testing (13, 14).

Alternatives to ic-MPT should preferably demonstrate the consistency of the content and quality of vaccine components, establish the presence or absence of correlates of vaccine efficacy, be sensitive to vaccine degradation, demonstrate reproducibility, and exhibit cost effectiveness, ease of accessibility, and transferability to other labs (13, 15). By these measures, neither of the alternatives to the ic-MPT currently under consideration, namely the pertussis serology potency test (PSPT) or the intranasal/aerosol challenge model, has been fully qualified as a suitable replacement for routine wP potency testing. A less explored alternative, proposed by Xing and colleagues, is the identification of pertussis biomarkers that can predict the outcome of protective adaptive immune responses—an approach that could be valuable in establishing alternate wP potency tests (12). Furthermore, new technical developments, such as protein microarrays, for uncovering immune responses triggered by wP vaccinations and identifying biomarkers of protection are invaluable resources. In this study, we employed a proteomic *B. pertussis* protein microarray after vaccination with optimal and suboptimal DTwP preparations to identify a serological "signature" that correlates with wP vaccine potency in mice.

## MATERIALS AND METHODS

### *B. pertussis* strains and culture conditions

*B. pertussis* strain 18323 was obtained from the American Type Culture Collection (code 9797). Glycerol stocks were plated for single colonies on Oxoid charcoal agar medium supplemented with 10% horse blood. The plates were incubated at 37°C for 48 h. For liquid cultures, single *B. pertussis* colonies were used to inoculate Stainer-Scholte broth supplemented with 1 mg/mL Heptakis (2,6-di-O-methyl)-β-cyclodextrin and 1× Stainer-Scholte supplement, according to Hulbert and Cotter (16), then incubated with aeration on a rotary shaker (200 rpm) for 16 h or until cultures reached mid-log phase. For intranasal challenge studies, mid-log phase cultures of *B. pertussis* were adjusted to an optical density ($OD_{600}$) of 0.1 (~$1 \times 10^8$ colony forming units (CFUs)/mL) by the addition of phosphate-buffered saline (PBS). Optical density was determined using a Genesys 10S UV-vis spectrophotometer (Thermo Fisher Scientific, Waltham, MA).

## DTwP vaccine formulations

A mock DTwP vaccine was prepared at lab scale at the University of Kansas using antigen bulks (diphtheria toxoid, DT; tetanus toxoid, TT; and thimerosal-inactivated wP) kindly provided by BioFarma (Bandung, Indonesia) based on formulation procedures described elsewhere for preparing mock wP-containing combination vaccines with aluminum-salt adjuvants (8). Individual 7.6× mock formulations of TT and DT antigens with aluminum-phosphate adjuvant (Adju-Phos, AP, purchased from InvivoGen, CA, USA) were prepared aseptically by combining calculated amounts of stock solution of each antigen with a stock AP to a final concentration of 114 Lf/mL TT, 304 Lf/mL DT, and 2.5 mg/mL of AP to a target pH of 7.0 and incubated at room temperature for 1 h. Then, the 7.6× TT and DT mock formulations were mixed in equal volumes with wP antigen and calculated volumes of 0.5 mM sodium phosphate, 150 mM NaCl, and pH 7.0 buffer to prepare 2× DTwP vaccine. Two times DTwP vaccine was diluted twofold in 0.5 mM sodium phosphate, 150 mM NaCl, and pH 7.0 buffer to prepare 1× DTwP vaccine (40 Lf/mL of DT, 15 Lf/mL TT, 24 opacity units (OU)/mL: wP, and 0.66 mg/mL AP) in stoppered 5 mL glass vials and stored upright at 2–8°C overnight before use.

## Heat treatment of DTwP vaccines

DTwP formulations (200 µL aliquots) were subjected to thermal stress incubation in water baths maintained at 60°C, 80°C, or 100°C for 60 min. Following incubation, tubes were placed on ice for 5 min, and aliquots were combined and stored at 2°C–8°C (subsequently referred to as 4°C for simplicity) until use.

## Mouse vaccinations

Animal studies were conducted in compliance with the Wadsworth Center's Institutional Animal Care and Use Committee. Female BALB/c mice aged 6–8 weeks at the start of experiments were obtained from Taconic Biosciences (Germantown, NY) and housed under conventional, specific-pathogen-free conditions. Mice ($n = 3–4$ per group) were vaccinated with DTwP as described above for a total of seven independent experiments. Immediately prior to use, vaccines were diluted in PBS (0.5 mM phosphate) to achieve the desired OU. The vaccines were administered to mice on study day 0 in a final volume of 250 µL via the intraperitoneal (IP) route for prime immunizations. For prime-boost immunization regimes, different sets of mice were vaccinated on days 0 and 21. Mice were weighed just prior to vaccination (day 0), then again on days 1 and 7 as an indirect measure of vaccine-induced morbidity. Blood was collected from mice via the submandibular vein on study days 21 and 30.

## *B. pertussis* intranasal challenge model

BALB/c mice (female, aged 4–5 weeks at the start of experiments) were obtained from Taconic Biosciences. Mice ($n = 4–6$ per group) were vaccinated with 1.2 OU/mouse DTwP in a 250 µL volume by IP injection on study day 0 (prime) or days 0 and 21 (prime/boost). Blood was routinely collected on days 21 and 30 by the submandibular route. On study day 35, mice were anesthetized with isoflurane and intranasally challenged with a 50 µL suspension of ~$5 \times 10^6$ CFUs of mid-log phase *B. pertussis* strain 18323. On study day 39 (4 days post-challenge), mice were euthanized by $CO_2$ asphyxiation, and lung tissues were dissected and homogenized twice at 5 m/s for 30 s in a Bead Mill 4 homogenizer (Fisher Scientific, Hampton, NH). Lung homogenates were serially diluted and plated on Oxoid charcoal agar supplemented with 10% horse blood. Plates were incubated at 37°C for 3–5 days, and *B. pertussis* CFUs were enumerated visually.

## Bacterial whole-cell ELISA

*B. pertussis* strain 18323 grown on Oxoid charcoal agar was collected by scraping and resuspension in PBS and adjusted to an $OD_{600}$ of 0.1. Aliquots (100 µL) of live bacterial

cells were added to each well of flat-bottomed Nunc Immulon 4HBX 96-well plates (Thermo Fisher Scientific, Waltham, MA), then placed overnight at room temperature in a biosafety cabinet with ventilation to allow evaporation. When plates were visibly dry, they were treated for 2 h with 200 µL of block buffer consisting of PBS with 0.05% Tween-20 (PBS-T) and 2% goat serum (Gibco, MD, USA). Serum samples were diluted 1:100, followed by threefold serial dilutions in block buffer across the plate. Pooled sera from DTwP-hyperimmunized mice served as a positive control, while pooled sera from mock (PBS) vaccinated mice served as a negative control. The plates were incubated for 1 h at room temperature, washed three times with PBS-T, and then overlaid for 30 min with horseradish peroxidase (HRP)-conjugated, goat anti-mouse IgG antibody (Southern Biotech, Birmingham, AL). Plates were washed four times, then developed for 4 min with SureBlue 3,3′,5,5′-tetramethylbenzidine (TMB; KPL, Gaithersburg, MD) followed by the addition of stop solution (1 M phosphoric acid). The absorbance at $OD_{450}$ for each plate was measured with the SpectraMax iD3 spectrophotometer equipped with Softmax Pro 7.1.0 software (Molecular Devices, San Jose, CA). The endpoint titer was defined as the reciprocal of the highest serum dilution with an absorbance reading three times above the background, with the background being defined as the average absorbance produced by wells with block buffer alone. Endpoint titers for each sample were transformed into geometric means and plotted using GraphPad Prism for visualization.

## Multiplex immunoassay

TT (Cayman Chemical, Ann Arbor, Michigan), diphtheria toxoid (CRM197; Cayman Chemical, Ann Arbor, MI), pertussis toxin (PT; Sigma-Aldrich, St. Louis, MO), 69 KDa pertactin (PRN; List Labs, Campbell, CA), filamentous hemagglutinin (FHA; Sigma-Aldrich), and adenylate cyclase toxin (List Labs, Campbell, CA) were each coupled to Luminex xMAP MagPlex-C microspheres of different spectral according to manufacturer's instructions (Luminex Corp, Austin, TX). Briefly, $1.25 \times 10^6$ microspheres were washed using activation buffer (0.1 M monosodium phosphate, pH 6.2) and activated by adding 50 mg/mL sulfo-NHS (N-hydroxysulfosuccinimide) and EDC (1-ethyl-3-[3-dimethylamino-propyl] carbodiimide hydrochloride; Thermo Fisher Scientific). Each bead region was then incubated with 6.25 µg of the respective antigen, yielding a final concentration of 6.25 µg antigen/$1 \times 10^6$ beads in coupling buffer (0.5 M 2-[N-morpholino] etha-nesulfonic acid, pH 5.0). Following wash steps, the resultant antigen-coupled beads were resuspended in storage buffer (PBS with 1% bovine serum albumin [BSA], 0.02% Tween-20, 0.05% azide, and pH 7.4) and stored at 4°C.

Each antigen-coupled bead was vortexed and sonicated for 10 seconds. Antigen-cou-pled beads were then pooled and combined at equal proportions for a final 1:50 dilution in Luminex PBN Buffer (PBS, 1% BSA, and 0.05% sodium azide). Thereafter, 50 µL of bead mixture was added to 50 µL of serum samples previously diluted to 1:100 in each well of a black opaque 96-well flat bottom plate (Greiner Bio-One). Plates were incubated for 1 h at room temperature in the dark with shaking at 600 rpm on a tabletop shaker (Thermo Fisher Scientific). After 1 h of incubation, samples were manually washed three times by dispensing 200 µL of Luminex wash buffer (PBS, 2% BSA, 0.02% Tween 20, 0.05% azide, and pH 7.5) into each well for 2 min with the plates mounted on a 96-well plate magnetic separator. After washing, samples were incubated with 100 µL of a 1:500 diluted phycoerythrin-tagged goat-anti mouse IgG-Fc (Southern Biotech) in the dark at room temperature (RT) with shaking at 600 rpm for 30 min. Plates were washed again three times as described above, and the resultant bead-coupled antigen-antibody complexes were resuspended in 100 µL Luminex wash buffer and incubated shaking (600 rpm) for 1 min at RT in the dark. The antigen-specific antibody binding in each sample was measured using a FlexMap 3D instrument (Luminex Corp.), with results reported as median fluorescence intensity (MFI). For each antigen, MFI values were normalized by dividing the MFI of each test sample by the average MFI from naive mice, generating an MFI index.

## Pertussis competition ELISA

Immulon 96-well ELISA plates were coated overnight at 4°C with 50 µL/well of DTwP vaccine that had not been subjected to thermal stress at 0.25 OU/mL. Coated plates were blocked for 2 h with 2% goat serum in PBS-T. Twofold serial dilutions of unstressed and thermally stressed DTwP (incubated prior at 40°C, 60°C, 80°C, or 100°C) were mixed with hyperimmune DTwP serum at concentrations equivalent to their relative 90% maximal effective concentration ($EC_{90}$). The DTwP-hyperimmune serum mixtures were incubated for 60 min at room temperature in a polyvinyl chloride (PVC) dilution plate and then applied to the microtiter plates coated prior with unstressed DTwP and incubated for 1 h for competition. Control wells contained only hyperimmune serum at concentrations required to achieve $EC_{90}$. Plates were then washed with PBS-T and probed with HRP-conjugated goat anti-mouse IgG for 30 min. The plates were washed again and developed with SureBlue TMB. Relative half maximal effective concentrations ($EC_{50}$) values were generated following the guidelines provided by SeaBaugh (17). Results were plotted to represent the percent of antibodies able to bind DTwP on the plate when compared to wells that contained serum alone without a competitor. $EC_{50}$ values were determined with a 4-PL sigmoidal curve fit using GraphPad Prism 9 (GraphPad Software, San Diego, CA).

## *B. pertussis* partial proteome microarray

A protein microarray with limited proteome coverage was developed for this study at Antigen Discovery, Inc. (ADI, Irvine, CA). Proteins included on the microarray were identified and selected as described (18). Briefly, proteins were selected using *in silico* prediction of antigenic targets. Proteins were selected based on a prioritization scheme that consisted of ranking of proteins by scores for protein features that predict surface localization and secretion. Methods used for scoring included (i) major antigen families identified in the literature search, (ii) prediction of transmembrane domains using TMHMM 2.0 (19), (iii) prediction of signal peptides using SignalP 5.0 (20), (iv) prediction of subcellular localization using PSORTb 3.0.2 (21) with priority from highest to lowest for: outer membrane, extracellular/secreted, plasma membrane, and periplasmic, (v) gene ontology word search (22) with the following search terms: flagellin, flagella, flagellar, Fli, fimbrial, membrane, secreted, adhesin, cell wall, surface, hemagglutinin, transport, cilia, pilus, pili, Pil, porin, holin, Ton, Omp, receptor, transfer protein, toxin, antigen, trigger factor, protease, lysin, and adhesin.

Proteome microarrays were fabricated by using a library of partial or complete open reading frames (ORFs) cloned into a T7 expression vector pXI that has been established at ADI. Briefly, the clone library was created through an *in vivo* recombination cloning process with PCR-amplified coding sequences from the *B. pertussis* Tahoma I strain, and a complementary linearized expressed vector transformed into chemically competent *Escherichia coli* cells was amplified by PCR and cloned into the pXI vector using a high-throughput PCR recombination cloning method as described in detail elsewhere (23). A total of 151 ORFs representing 127 unique pertussis genes were cloned and sequenced (Retrogen, Inc., San Diego, CA), and the results matched the correct target for all clones. Proteins were expressed using an *E. coli in vitro* transcription and translation (IVTT) system (Rapid Translation System, BiotechRabbit, Berlin, Germany). Each expressed protein included a 5′ polyhistidine epitope tag and a 3′ hemagglutinin (HA) epitope tag. After expressing the proteins according to the manufacturer's instructions, translated proteins were printed onto nitrocellulose-coated glass AVID slides (Grace Bio-Labs, Inc., Bend, OR) using an OmniGrid accent robotic microarray printer (Digilabs, Inc., Marlborough, MA). Each slide contained 16 nitrocellulose pads on which the full array was printed (this allowed 16 samples to be probed per slide using sealed chambers that isolated the arrays). In addition to the targeted proteins, IVTT reactions without expression insert were included and spotted in replicates on each subarray of each pad. These "IVTT controls" served as a normalization factor for array-to-array variation. Microarray chip printing and protein expression were quality checked by

probing random slides with anti-His and anti-HA monoclonal antibodies with fluorescent labeling.

Mouse serum samples were diluted to a 1:100 final concentration in a 1.5 mg/mL *E. coli* DH5α lysate solution (Antigen Discovery, Inc., Irvine, CA) in Surmodics Assay Diluent (Surmodics, Inc., Eden Prairie, MN) and incubated at room temperature for 30 min before applying samples to arrays. Arrays were incubated overnight at 4°C with agitation, washed three times with Tris-buffered saline (0.05% Tween 20; Thermo Scientific, Cat# J77500K8, diluted 20× in molecular grade water), and incubated with Cy5-conjugated anti-mouse IgG (Cat# 115–175-071, Jackson ImmunoResearch, West Grove, PA) diluted 1:200 in Assay Diluent. Washed and dried microarray slides were scanned using a GenePix 4300A high-resolution microarray scanner (Molecular Devices, Sunnyvale, CA), and signals were quantified using Mapix software (Innopsys, Carbonne, France). All further data processing was performed in R (http://www.R-project.org). Data were normalized by first transforming raw values using the base 2 logarithm, then subtracting the median signal intensity (SI) of the IVTT control spots for each sample. This procedure provides a relative measure of the specific antibody binding vs the nonspecific antibody binding to the IVTT expression system (i.e., signal-to-noise ratio). With the normalized data, a value of 0.0 means that the intensity is no different than that of the IVTT controls, and a value of 1.0 and every unit increase thereafter indicates a doubling with respect to IVTT control spots.

## Statistical analysis

For data sets involving analysis of only two treatment groups (i.e., $DTwP_4$ and $DTwP_{100}$ only), differences between groups were assessed using a standard unpaired *t*-test when the population had a Gaussian distribution with equal variances. On the other hand, a non-parametric Mann-Whitney test was used where populations did not have a Gaussian distribution. In instances where variances between groups were unequal but the data set was normally distributed, an unequal variance (Welch) *t*-test was performed. For normally distributed data sets involving more than two groups (i.e., $DTwP_4$, $DTwP_{60}$, $DTwP_{80}$, and $DTwP_{100}$), we performed a one-way Analysis of variance (ANOVA) when SDs between groups were equal and Brown-Forsythe and Welch ANOVA tests when groups did not have equal SDs. In all cases, differences between groups were significant when *P* value ≤ 0.05. For microarray data, nonparametric methods were used for group comparisons. Discrimination of antibodies between vaccine temperature stress conditions was assessed using the area under the receiver operator characteristics curve (AUC). Significance was assessed using Wilcoxon's rank sum tests. To assess the effect of bleed time point and dose regimen (prime-only or prime-boost), multivariable linear regression models were fit to the data. All *P* values were adjusted for the false discovery rate as described by Benjamini and Hochberg (24).

## RESULTS

### DTwP dosing regimen in mice

Before embarking on potency studies, we conducted a dose-response analysis of DTwP in mice to define the limiting amount of vaccine in our model. Groups of animals received a single intraperitoneal injection of DTwP across a range of OU per mouse (range 0.01–9 OU/mouse) or PBS for naive controls. We employed DTwP for the sake of these studies for reasons relating to the availability of bulks and its status as the base formulation for combination vaccines containing wP. Serum samples were collected 30 days later, and *B. pertussis*-specific serum IgG responses were assessed using whole-cell *B. pertussis* strain ELISA, as described in the Materials and Methods. In addition, multiplexed immunoassays (MIA) were performed with three different *B. pertussis* antigens: PRN, PT, and FHA.

The geometric mean titer (GMT) of *B. pertussis*-specific IgG increased in a dose-dependent manner between 0.01–3 OU, then plateaued (Fig. 1A). GMTs ranged from

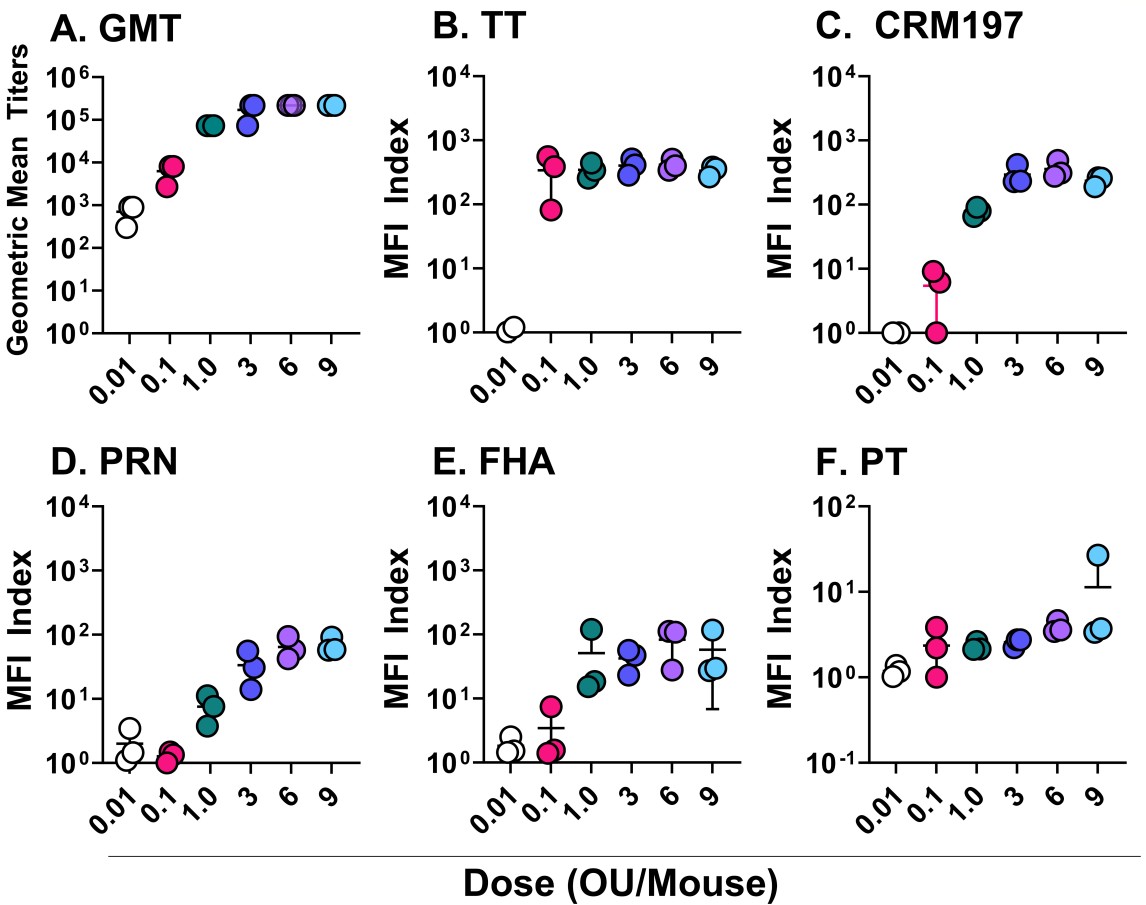

**FIG 1** Dose-dependent antibody response to DTwP vaccination. Groups of mice (*n* = 3) received a single IP dose of either DTwP or PBS on day 0, with doses ranging from 0.01 OU/mouse to 9 OU/mouse. Sera were collected 30 days post-vaccination. Total wP-specific antibody responses were measured by wP ELISA (A), while antigen-specific antibody responses were assessed by MIA (B–F).

300 to 900 in mice that received 0.01 OU to as high as 218,700 in some mice that received 3 OU. Naive animals did not generate a detectable response (data not shown). In terms of specific *B. pertussis* antigens, MFI values for PRN and FHA increased in a dose-dependent manner, plateauing at 3 OU and 1 OU, respectively (Fig. 1D and E). MFI values for PT were low at all doses tested, suggesting that there were insufficient levels of pertussis toxin in the DTwP formulation to induce an immune response in this model (Fig. 1F). The single dose of DTwP also elicited high titers of serum IgG against both TT and CRM-197 that saturated at 0.1 OU and 1 OU corresponding to (0.17 and 1.7 Lf/mL) for CRM-197 and (0.0625 and 0.625 Lf/mL) for TT, respectively (Fig. 1B and C). These results demonstrate that the limiting dose of wP is between 1 OU and 3 OU, an observation consistent with the literature (25). We chose 1.2 OU (1/10 human dose) for subsequent mouse vaccinations.

## Immunoreactivity of DTwP declines following thermal stress

The World Health Organization (WHO) uses an accelerated degradation test to assess the potency of wP vaccines (26). To investigate whether temperature-dependent loss of conformational integrity of DTwP antigens could be detected *in vitro*, we developed a qualitative competition ELISA that we refer to as PetCoE (pertussis competition ELISA; Fig. 2A). In short, soluble DTwP formulations (control or thermally stressed) were evaluated for their ability to inhibit pooled hyperimmune mouse sera raised against

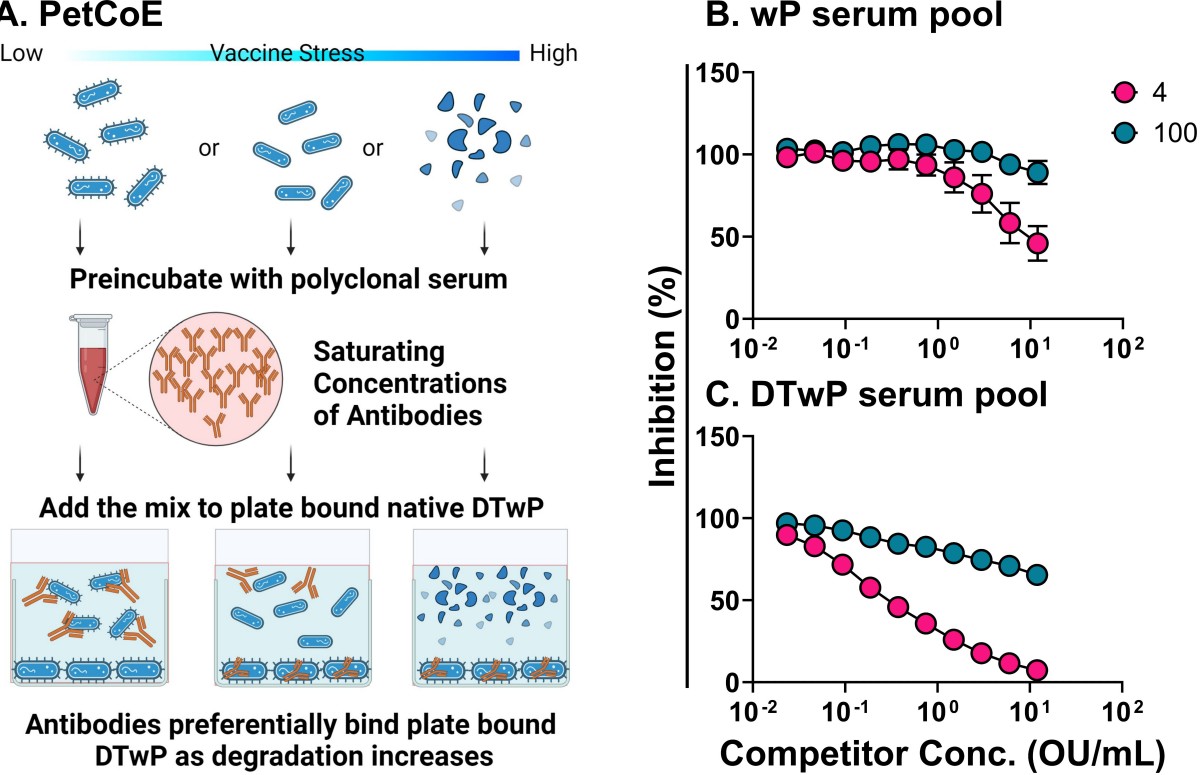

**FIG 2** Analysis of thermally denatured DTwP by PetCoE. Twofold serial dilutions of DTwP (previously incubated at 4°C or 100°C for 1 h) were mixed with anti-wP (B) or -DTwP (C) hyperimmune serum at a fixed dilution equivalent to 90% maximal binding. The DTwP-serum mixtures were incubated for 1 h at room temperature before being applied to microtiter plates coated with DTwP. The plates were washed and probed with HRP-conjugated goat anti-mouse IgG followed by SureBlue TMB. Each symbol represents the mean ± SD of at least three separate experiments. Error bars in Fig. 2C are not visible around the data points due to the low variability of the assay.

either bulk wP or DTwP drug product from binding to native DTwP adsorbed to 96-well microtiter plates (Fig. 2B and C). Using this assay, we found that DTwP preparations stored at 4°C (DTwP$_4$) had EC$_{50}$ values of ~0.5 OU/mL using in-house generated DTwP or wP hyperimmune sera. In contrast, DTwP aliquots subjected to 100°C for 60 min (DTwP$_{100}$) lost the ability to compete with *B. pertussis* DTwP or wP hyperimmune serum (Fig. 2), suggesting that the integrity of the DTwP$_{100}$ preparations had been compromised. The wP-only hyperimmune serum pool curve confirmed that the observed shift in binding at DTwP$_{100}$ was largely due to pertussis-specific antigens and not DT or TT (Fig. 2B and C).

**Thermal stress reduces DTwP potency in a mouse model of *B. pertussis* intranasal challenge**

Having established that thermal stress results in an apparent reduction of wP reactivity in the PetCoE assay, we next investigated whether there was a corresponding impact on vaccine potency *in vivo*. Groups of mice were vaccinated intraperitoneally on study day 0 with 1.2 OU of control (4°C; DTwP$_4$) or stressed (100°C; DTwP$_{100}$). Serum samples were collected on study days 21 and 30. Mice were challenged intranasally with *B. pertussis* 18323 (5 × 10$^6$ CFU) on day 35. On day 39 (4 days post-challenge), the mice were euthanized, and lung homogenates were assessed for *B. pertussis* CFUs as a proxy for bacterial colonization.

Lung homogenates from sham-vaccinated mice collected 4 days after intranasal challenge contained an average of ~10$^7$ CFUs of *B. pertussis* 18323 (Fig. 3A). By comparison, lung homogenates from DTwP$_4$-vaccinated animals had an average of ~10$^4$ CFUs, which corresponds to a 1,000-fold reduction in bacterial burden as a result of

vaccination. Mice that received DTwP$_{100}$ had an average bacterial load of ~10$^5$ CFUs, corresponding to a 100-fold reduction in bacterial burden relative to sham-vaccinated mice but 10-fold more than DTwP$_4$-vaccinated mice (Fig. 3A). Thus, the potency of DTwP$_{100}$ is reduced ~10-fold *in vivo*.

In terms of vaccine immunogenicity, total *B. pertussis* antibody titers, as determined by ELISA, were significantly elevated in DTwP$_4$ and DTwP$_{100}$ vaccinated animals on days 21 and 30, as compared to sham vaccinated controls (Fig. 3B). However, end point titers between the DTwP$_4$ and DTwP$_{100}$ groups of mice were not significantly different, demonstrating that total *B. pertussis* IgG levels do not discriminate between potent and subpotent vaccines in this model (Fig. 3B).

The picture was markedly different when we examined antigen-specific antibody titers. Specifically, DTwP$_4$ vaccination elicited TT- and CRM197-specific serum IgG titers that were >100-fold increased over baseline (i.e., naïve serum) on days 21 and 30 (Fig. 3C and D). CRM197-specific antibody titers were similar in DTwP$_{100}$ vaccinated animals, whereas TT antibody titers were reduced >60-fold (Fig. 3C and D). In terms of *B. pertussis* antigens, DTwP$_4$ vaccination elicited FHA- and PRN-specific titers on days 21 and 30 that were ~10-fold and ~100-fold over background, respectively (Fig. 3E and F). FHA antibody titers were similar between DTwP$_4$ and DTwP$_{100}$ vaccinated animals, whereas

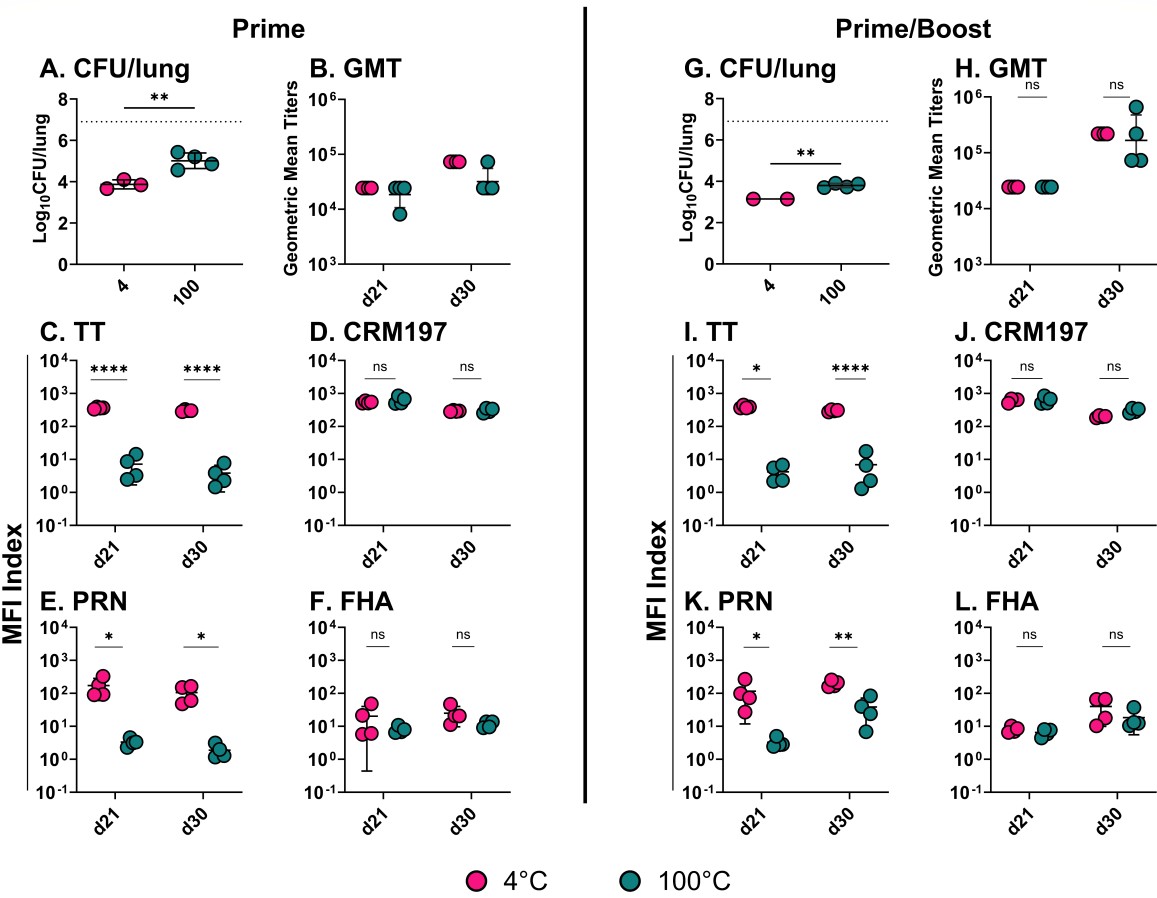

**FIG 3** *B. pertussis* intranasal challenge indicates DTwP potency loss following thermal stress. Groups of mice (*n* = 4/group) were intraperitoneally vaccinated with either a prime dose on day 0 or a day 0 prime dose followed by a booster dose on day 21. Each dose consisted of 1.2 OU/mouse of DTwP incubated at 4°C or 100°C. Sera were collected on days 21 and 30 with an intranasal challenge with *B. pertussis* 18,323 (5 × 10$^6$ CFU) on day 35. Lungs were harvested 4 days later so that the resulting CFUs/lung could be compared between dosing strategies (A and C). The average CFU/lung recovered from sham animals is represented by a dashed line. Sera samples were similarly compared using whole bacterial ELISAs to generate anti-pertussis GMT (B and D) and MIA to provide antigen-specific titers (E–L). Statistical analysis was performed using unpaired t test, with or without Welch's correction, or nonparametric Mann-Whitney test. One of the two experiments is shown.

PRN antibody titers were reduced by >100-fold. Thus, immunoreactivity with TT and PRN may have the potential to serve as proxies for DTwP thermal stress.

We repeated the intranasal challenge study with the addition of a booster vaccination on study day 21 to determine whether the sub-potency of DTwP$_{100}$ could be restored upon additional antigen exposure. While the prime/boost regimen did result in reduced *B. pertussis* colonization following intranasal challenge in both the DTwP$_4$ and DTwP$_{100}$ vaccinated groups of mice, as compared to their prime-only counterparts, DTwP$_{100}$ remained inferior to DTwP$_4$ (Fig. 3G). Similarly, DTwP$_{100}$ vaccinated mice had significantly reduced PRN-specific antibody titers (as well as TT antibodies) compared to DTwP$_4$-vaccinated mice, suggesting that thermal stress irrevocably impacts components of DTwP (Fig. 3I and K). Total antibody titers, as well as CRM197- and FHA-specific antibody titers, were not significantly different between the groups (Fig. 3H, J and L).

## Refinement of the DTwP accelerated decay assays

We next assessed the potency of DTwP following a lower range of stress temperatures (40°C, 60°C, and 80°C for 60 min). In the PetCoE assay, inhibition (EC$_{50}$) curves associated with DTwP$_{40}$ and DTwP$_{60}$ were superimposable with DTwP$_4$, indicating that neither of those temperatures adversely affected vaccine integrity after incubation for 1 h (Fig. 4A). In contrast, DTwP$_{80}$ resembled the DTwP$_{100}$ profile in that it had reduced inhibitory activity in PetCoE (Fig. 4A).

The same thermally stressed DTwP formulations were evaluated *in vivo*. Specifically, groups of mice were vaccinated on study days 0 (prime) and 21 (boost) with 1.2 OU of DTwP and then challenged intranasally with *B. pertussis* 18323 (5 × 10$^6$ CFU) on day 35, as described above. As was seen with previous studies, no differences were noted in geometric mean titers across all groups (Fig. 4C). DTwP$_4$- and DTwP$_{60}$-vaccinated mice had significantly reduced *B. pertussis* CFUs (>100-fold) in lung homogenates, relative to the sham-vaccinated mice (Fig. 4B), indicating that the relative potency of DTwP$_{60}$ was largely unaffected. In contrast, bacterial loads in DTwP$_{80}$ and DTwP$_{100}$ were significantly higher than DTwP$_4$-vaccinated animals, demonstrating that, in our model, 80°C and 100°C stress conditions resulted in a ~10-fold reduction in DTwP potency even in prime/boost vaccination regimen. Analysis of serum samples collected on day 30 just prior to *B. pertussis* challenge revealed a trend toward lower PRN-specific IgG titers as a function of thermal stress, as well as a near complete loss of TT antibodies in mice that received DTwP$_{100}$ (Fig. S1). Antibody levels to FHA and CRM197 were unchanged across the four

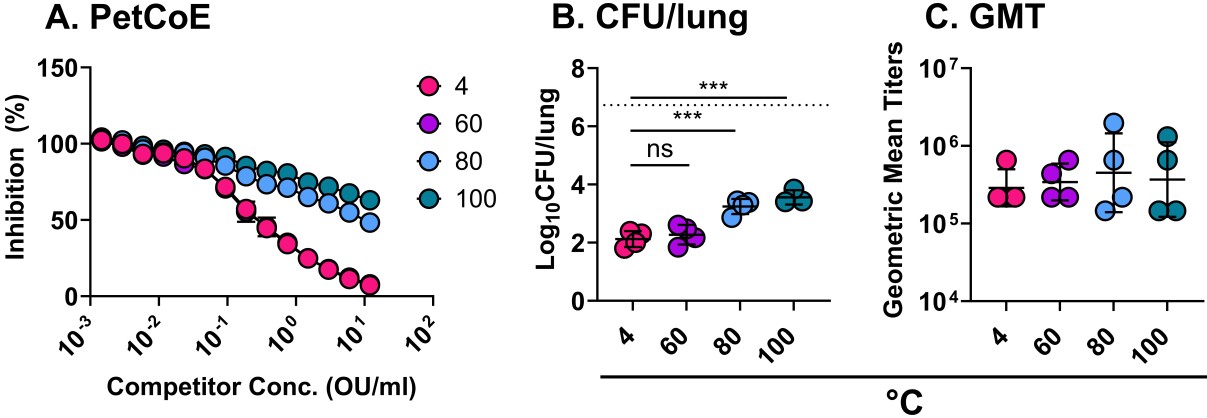

**FIG 4** DTwP potency assessment following temperature stress. DTwP samples were incubated at different temperatures and assessed by PetCoE (A). The same samples were used to intraperitoneally vaccinate mice (*n* = 4) on days 0 and 21 with 1.2 OU/mouse. Sera were collected on day 30 followed by an intranasal challenge with *B. pertussis* 18,323 (5 × 10$^6$ CFU) on day 35. Lungs were harvested 4 days post-challenge, homogenized, and plated so that CFUs/lung could be quantified (B). The average CFU/lung recovered from sham animals is represented by a dashed line. Sera were utilized in whole-cell ELISAs to determine anti-pertussis GMT (C). The mean ± SD is shown for PetCoE, which was performed in triplicate. Statistical analysis for CFUs and GMTs was performed using one-way ANOVA. One of two experiments is shown.

groups of animals ($DTwP_4$, $DTwP_{60}$, $DTwP_{80}$, and $DTwP_{100}$). Thus, DTwP potency declines with increasing temperature stress and may be accompanied by measurable changes in antigen-specific antibody profiles.

## Proteome analysis of *B. pertussis* antigens recognized by wP immune sera

We sought to investigate on a larger scale whether changes in DTwP potency as a result of thermal stress are accompanied by distinct alterations in antigenicity profiles. In other words, is there a serological signature associated with wP vaccination in mice, and do changes in that signature correlate with vaccine potency? Before embarking on such a study, it was necessary to define reactive antigens associated with DTwP-vaccinated mice inclusive of all experimental groups ($DTwP_4$, $DTwP_{60}$, $DTwP_{80}$, and $DTwP_{100}$). For this purpose, a total of 134 pre-challenge serum samples (day 30) collected from $DTwP_4$, $DTwP_{60}$, $DTwP_{80}$, and $DTwP_{100}$ vaccinated mice were evaluated using a *B. pertussis* Tohama I limited proteome array (4% genome coverage) consisting of 151 spots encompassing 127 unique pertussis antigens. Also included on the array were two purified recombinant proteins, fimbriae 2/3 (GHA) and filamentous hemagglutinin (fhaB). A reactive antigen was defined as having an IgG SI greater than twofold over background (seropositive) in at least 10% of the mice in the study (i.e., ≥10% seroprevalence). For purified recombinant proteins, the seropositivity threshold was a raw fluorescence SI of 500.

The analysis identified a total of 34 reactive IVTT antigens or antigen fragments with ≥10% seroprevalence (Table S1). Global median SI across the 34 antigens ranged from 0.09 to 6.78 with seroprevalence ranging from 12% to 100%. The top eight antigens stood out as having notably higher global median reactivities (>2) and seroprevalence values (>80%) than the rest of the list. The eight antigens were enriched in membrane-associated proteins, including three known or putative outer membrane proteins (BP3755, BP0943, and BP0840), two lipoproteins (BP2992 and BP3342), the phg autotransporter (BP1767), and two fragments of the Bordetella intermediate protein A (BipA; BP1112). The remaining 26 antigens had seroprevalence values ranging from 12% to 58% and were enriched in secreted or membrane-associated proteins (Table S1). The purified recombinant protein FhaB was also serorecognized in 49% of vaccinated animals. Thus, within the DTwP collection of vaccinated mice, we identified 34 reactive *B. pertussis* antigens predominantly associated with the bacterial envelope.

## A distinct antibody profile associated with $DTwP_4$ vaccination

To identify antigens specifically associated with $DTwP_4$, serum samples from 51 BALB/c mice vaccinated with $DTwP_4$ on day 0 (prime) or days 0 and 21 (boost) were subjected to the *B. pertussis* proteome array, alongside 44 serum samples from sham-vaccinated mice. The array identified 30 immunoreactive IVTT *B. pertussis* antigens from the list of 34 (Table 1; Fig. S2). When rank ordered based on AUC and statistical significance, among the top hits were Pal (BP3342; $P < 8.8 \times 10^{-6}$), OmpA (BP0943; $P < 8.8 \times 10^{-6}$), and BipA (BP1112; $P < 8.0 \times 10^{-4}$). Other reactive *B. pertussis* antigens include a peptidase (BP1721; $P < 1.3 \times 10^{-4}$) and the phg autotransporter (BP1767; $P < 5.1 \times 10^{-4}$). Twenty of the antigens are reportedly reactive with convalescent serum and/or nasopharyngeal wash (NPW) from baboons and are essential for *B. pertussis* colonization or survival *in vitro* (27).

Thus, we established a preliminary *B. pertussis* serological profile of DTwP vaccination in mice with considerable antigen overlap with sera and NPW from convalescent non-human primates. Moreover, when comparing mice vaccinated once or twice with $DTwP_{60}$, $DTwP_{80}$, or $DTwP_{100}$ to sham-vaccinated mice, similar counts of significantly differential antigens were found, all with substantial overlap (Tables S4 to S9; Fig. S3).

## Differential antibody profiles associated with wP potency

In an effort to identify *B. pertussis* antigens whose immunoreactivity correlates with changes in wP potency, sera from mice vaccinated once or twice with $DTwP_{60}$, $DTwP_{80}$,

**TABLE 1** *B. pertussis* antigens recognized by mouse wP immune sera

| Bp id[a,b] | UniProt[c] | Description | Loc[d] |
|---|---|---|---|
| **3342** | Q7VU04 | Peptidoglycan-associated lipoprotein (Pal) | O |
| **2992** | Q7VUT2 | Putative lipoprotein (Pcp) | O |
| **0943** | Q7VZG6 | Outer membrane protein A (OmpA) | O |
| **0840** | Q04064 | Outer membrane porin protein precursor (Omp) | O |
| 3755 | Q7VT02 | Putative outer membrane protein (OmpW) | O |
| 1721 | Q7VXN0 | Putative peptidase | O |
| 2667 | Q7VVJ2 | Adhesin (FhaS) | na |
| 2235 | Q79GR8 | Putative type III secretion protein (BscC) | O |
| 1767 | Q79GU5 | Autotransporter phg | O |
| 1112 | Q7VZ27 | Bordetella intermediate protein A (BipA) | O |
| 3655 | Q7VT95 | Penicillin-binding protein 1A | I |
| 3366 | Q7VTY1 | Putative phage tail protein | na |
| 1201 | Q79G × 8 | Tracheal colonization factor precursor | O |
| 2497 | Q7VVY4 | Zinc protease | na |
| 2077 | Q7W9J8 | Efflux system outer membrane component | O |
| 1251 | Q7VYQ9 | Putative toxin | S |
| 3494 | Q45340 | Serum resistance protein (BrkA) | S |
| 1382 | Q7VYG0 | Flagellar hook-associated protein 1 | S |
| 1378 | Q7VYG4 | Flagellar basal-body rod protein FlgG | I and O |
| 2851 | Q7VV52 | Outer membrane porin protein precursor | O |
| 0760 | P0DKX7 | Bifunctional hemolysin-adenylate cyclase precursor | S |
| 1884 | P35077 | FHA transporter protein FhaC | O |
| 0184 | Q7W0F3 | Putative exported protein | I |
| 1879 | P12255 | FHA | O |
| 2315 | Q79GN7 | Autotransporter Vag8 | O |
| 1401 | Q7VYE7 | Flagellar assembly protein FliH | C |
| 2566 | Q7VVS1 | Exported endonuclease | na |
| 2434 | Q7VW38 | Probable periplasmic serine endoprotease DegP-like | P |
| 0456 | Q7VSG8 | Heme receptor | O |
| 1189 | Q7VYW4 | Lipoprotein | na |

[a]Bold indicates proteins identified by Gregg et al. (2023) as being essential for survival *in vitro* and *in vivo* in a baboon model for *B. pertussis* colonization identified by convalescent serum or NPW (27).
[b]Underline indicates *B. pertussis* antigens identified by Gregg et al. (2023) detected in convalescent baboon serum or NPW (27).
[c]UniProt accession (28).
[d]Abbreviations indicate known or proposed cellular location of protein: O, outer membrane; I, inner membrane; P, periplasm; C, cytoplasm; S, secreted; na, not available.

or DTwP$_{100}$ were subject to the Tohama I limited proteome array as indicated above. In each case, we examined the reactivity of DTwP. When we plotted median reactivity values as a heat map (Fig. 5) for each of the top 30 *B. pertussis* antigens across the DTwP vaccination groups, we identified three broad reactivity antigen categories with statistical significance achieved primarily when comparing DTwP$_4$ to DTwP$_{60}$, DTwP$_{80}$ or DTwP$_{100}$ (Tables S10 to S15).

### Category 1-unchanged

This category was defined as having no significant change in antigen reactivity as a function of DTwP potency. The majority (23) of the 30 antigens fell within this group. For example, antibody reactivity to OmpW (**BP3755**) and OmpA (**BP0943**) was consistently high (>5) across all the different groups of vaccinated mice, irrespective of whether the animals had received potent or subpotent DTwP. Other examples within this category are putative lipoprotein BP2992 and FHA (BP1879; Fig. 6).

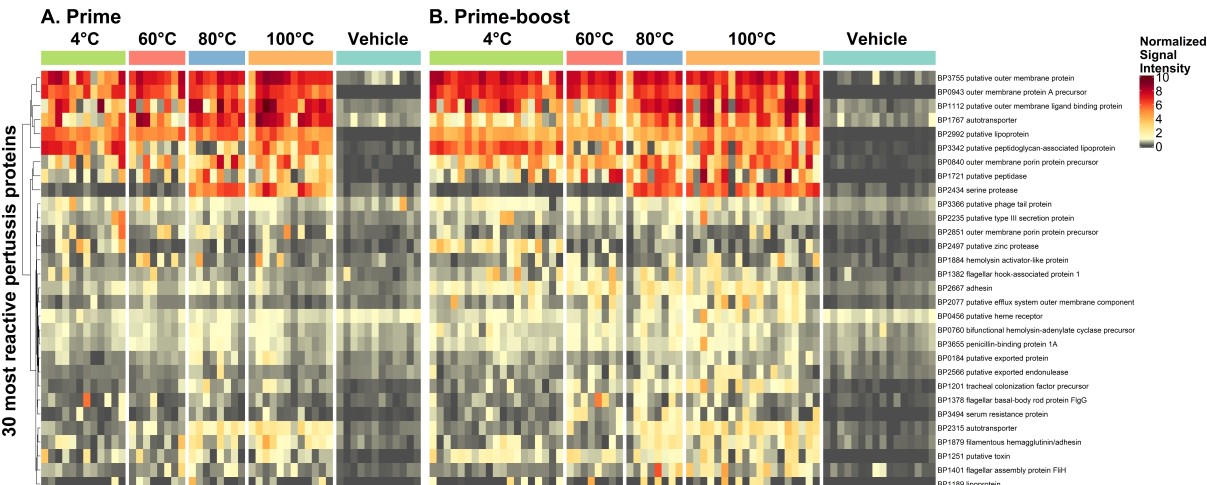

**FIG 5** Antibody reactivity profile of individual mice to 30 *B. pertussis* proteins. The heatmaps show the antibody binding normalized signal intensity on a colorimetric scale for each mouse sample (columns) and each of 30 reactive antigens (rows) according to our criteria of antigens with a seropositive response in at least 10% of DTwP-immunized mice. Normalized signal intensity represents the $\log_2$ signal-to-noise ratio of the specific antibody binding to the array and expression system background. Among the 34 identified reactive antigens, 4 that represented multiple fragments of the same gene were omitted. Mouse samples were ordered by temperature stress groups and stratified by prime (A) and prime-boost (B) regimens. The vehicle is the group of mice given the sham vaccine.

### Category 2-declined

This category was defined as antibody reactivity observed in sera from mice vaccinated with $DTwP_4$ but lower in $DTwP_{60}$, $DTwP_{80}$ and/or $DTwP_{100}$ vaccinated mice. Two *B. pertussis* antigens fell within this category: Pal (**BP3342**) in $DTwP_{80}$ and a zinc protease (**BP2497**) in $DTwP_{100}$. In the case of Pal, the antibody reactivity profile was "U" shaped, with similar reactivity in mice that received $DTwP_4$ and $DTwP_{100}$ but lower reactivity in those that received $DTwP_{60}$ and $DTwP_{80}$. In the case of BP2497, it was low at both $DTwP_{80}$ and $DTwP_{100}$ treatments (Fig. 6).

### Category 3-enhanced

This category was defined as weak or non-reactive antibodies in sera from $DTwP_4$- and $DTwP_{60}$- vaccinated mice but reactive in $DTwP_{80}$ and/or $DTwP_{100}$-vaccinated mice. Five antigens fell into this group with **BP2315** (autotransporter) and **BP2434** observed consistently in prime and prime/boost $DTwP_{80}$- and $DTwP_{100}$-vaccinated mice. The other antigens in this category include **BP1112** (BipA), **BP1767** (autotransporter), and **BP0840** (outer membrane porin precursor; Fig. 6).

### Multivariable modeling shows the effects of temperature stress, booster dose, and time point but not challenge

Using linear models on antibody levels of the 34 reactive IVTT proteins and 2 purified protein spots adjusting for temperature stress conditions (modeled as a continuous variable), bleed time point, and dosing regimen, general trends could be observed (Table S16). The strongest effect was temperature stress, where 16 of the 36 antigens were significantly associated with a change in antibody levels. Notably, PAL and the zinc protease BP2497 were the only two proteins significantly associated with a decrease in antibody levels as stress temperatures increased. Administration of the booster dose again showed nine antigens associated with a change in antibody levels, with only one, the autotransporter BP1767, showing a decrease. Additionally, the bleed time point had an effect on antibody levels, with nine antigens associated with higher antibody levels at day 30 vs day 21 and BP1767 associated with lower antibody levels. However,

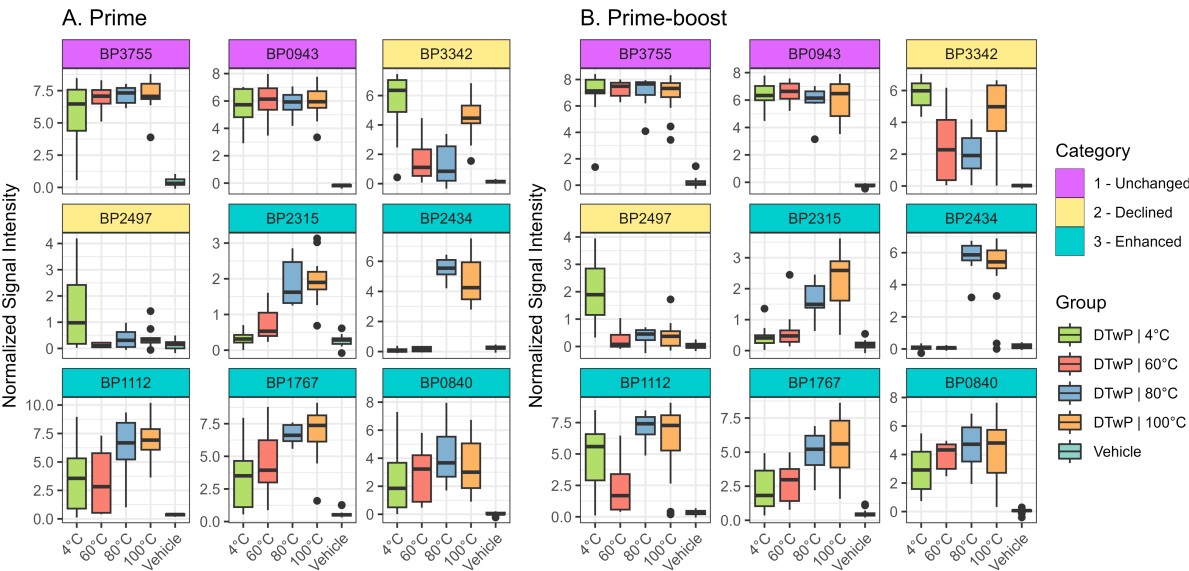

**FIG 6** Differential antibody levels between vaccine temperature stress conditions represent three categories of antibody binding profiles. The boxplots show antibody levels to select antigens representing three categories of antibody profiles associated with vaccine temperature stress: (i) unchanged (magenta headers); (ii) declined (yellow headers); and (iii) enhanced (teal headers). Each antigen profile is shown for prime (A) and prime-boost (B) regimens. The vehicle is the group of mice given the sham vaccine. Boxes represent the interquartile range (IQR), and bars represent the median. Upper and lower whiskers represent the furthest values within 1.5× the IQR from upper and lower quartiles, respectively. All data outside of this range are plotted individually as outlying points.

there were only five antigens with differential antibody levels between day 30 and the post-intranasal challenge bleed at day 39, with two showing decreases (including BP1767) and the remaining three showing increases ($P$ values ranged from $P = 0.02$ to $P = 0.04$).

## DISCUSSION

Having a methodology to readily assess the potency and stability of wP is of paramount importance with the demand for increasingly complex combination vaccines in LMICs as part of routine pediatric healthcare (8, 29). The Kendrick assay (ic-MPT) was developed in the 1950s, and to this day, remains the standard for potency determination and batch release testing of wP vaccines, despite widespread recognition that the assay itself is problematic to the point of being a liability (12, 25, 26). Several alternatives to the ic-MPT have been considered over the past two decades, including an aerosol challenge model (30–32) and the PSPT (10). The aerosol challenge model demonstrated good parallelism with the Kendrick assay and correlated with vaccine efficacy in children. However, implementing the respiratory challenge assay in a standardized fashion seems unlikely considering the expertise and resources required to generate uniform *B. pertussis* aerosols and the collection of lung tissues from large numbers of animals. The PSPT is technically simpler but suffers from the fact that it lacks functional antibody assessment (10). In addition, a recent initiative involving wP manufacturers demonstrated that putting PSPT into practice is not trivial (11, 13).

In this report, we used a limited *B. pertussis* proteome array in an effort to identify bacterial antigens whose antibody reactivity profiles correlate with potent and subpotent wP formulations. A prerequisite for such an investigation was the establishment of a forced degradation (FD) protocol to reproducibly render wP subpotent in the context of DTwP (15, 33). While the potency of wP is influenced by a multitude of extrinsic factors (26, 34), we chose thermal stress because it is the method recommended by the WHO for the Kendrick assay (26, 34). Thermal stress was also used by the PSPT consortium to deliberately alter wP potency (13). The choice of 40℃, 60℃, 80℃, and

100°C for 60 min is arguably extreme relative to WHO's recommendations (i.e., 43°C–45°C for 21 days); however, these stress conditions ensured stability indications could be detected in this work, and other groups have also resorted to similar modifications of the WHO protocol, albeit for aP-containing drug products (35).

Ultimately, our FD protocol proved robust and reproducible, *in vitro* and *in vivo*. In the PetCoE assay, for example, we observed a stepwise reduction in competitive inhibition of pertussis-specific hyper-immune sera as a function of thermal stress (i.e., 4°C, 60°C > 80°C > 100°C), indicating that the integrity of the DTwP preparations had indeed been compromised. The same pattern of attenuation was observed in the mouse vaccination/*B. pertussis* intranasal challenge model, where recoverable CFUs from lung tissues increased concomitantly with wP incubation temperature. It is notable that even after incubation at 100°C for 60 min, wP retained a significant degree of efficacy, as evidenced by the fact that mice that received $DTwP_{100}$ had >10-fold lower bacterial burdens following *B. pertussis* challenge as compared to mock vaccinated animals. This observation is consistent with there being both heat-labile and heat-stable antigens associated with wP. The most conspicuous heat-stable antigen is lipo-oligosaccharide (LOS), a major component of the *B. pertussis* outer membrane and a known target of protective antibodies (36, 37). We have confirmed by ELISA that DTwP vaccination elicits a strong antibody response to LOS (Y. Adewunmi, unpublished results).

A Luminex-based MIA with a handful of DTwP-specific antigens (PRN, FHA, PT, TT, and CRM197) proved a useful indicator of DTwP temperature-induced stress. Most notably, TT-specific antibody titers were reduced ~100-fold in mice vaccinated with $DTwP_{100}$, as compared to $DTwP_4$. Other groups have noted the temperature sensitivity of TT in the context of different combination vaccines (14). We also observed a temperature-dependent decline in PRN-specific antibody titers, suggesting that PRN may possibly serve as a stability-indicating antigen. Indeed, we are not the first to consider this possibility. Szeto and colleagues proposed the use of an antigenicity (sandwich) ELISA to assess the integrity (potency) of PRN within different aP-based combination vaccines (35). The utility of PRN as a stability-indicating antigen may depend in part on the choice of recombinant PRN used in the assay. PRN (UniProt P14283) is expressed as a large polypeptide (910 amino acids) that is processed into a transmembrane translocator (residues 632–910) and an extracellular beta-helix protein (residues 35–631; surface antigen P69) consisting of potentially dozens of surface-accessible B cell epitopes that may be prone to conformational alteration (38, 39).

In our study, total *B. pertussis* antibody titers, as determined by whole-cell ELISA, did not reflect changes in wP potency in the *B. pertussis* intranasal mouse challenge model. As a case in point, *B. pertussis* antibody titers were virtually identical in mice vaccinated with $DTwP_4$ and $DTwP_{100}$, even though the $DTwP_{100}$ was deficient in their ability to clear *B. pertussis* from lung tissues. This outcome was not unexpected, considering that LOS, a highly heat-resistant major component of the outer membrane of gram-negative bacteria, likely dominates the wP ELISA results and masks changes associated with specific protein antigens and other virulence factors important for protection which may have been heat labile (14). As such, wP ELISAs do not distinguish between functional and non-functional antibodies.

In the original PSPT study by Van der Ark, total wP antibody concentrations correlated with protection in mice. However, in our study, total wP antibody did not correlate with bacterial clearance/protection in mice lungs. This apparent discrepancy between our study and the PSPT may be explained by fundamental issues related to vaccine-induced antibody quantity vs quality. In the PSPT study, van der Ark and colleagues correlated the survival of mice in the ic-MPT assay to pertussis-specific serum antibody concentrations generated by vaccination with unique final drug products, each exhibiting different potency levels. However, based on our reading, the van der Ark studies never subjected wP samples to FD (40), which is an essential factor to consider when developing methods that define vaccine potency. By not including FD in their studies, their method does not account for suboptimal vaccines that are still capable of eliciting robust

but non-functional antibody responses, which would have played a role in defining their PSPT potency cutoff. As such, though the PSPT was successful in establishing a correlation between pertussis-specific antibody concentrations (based on ELISA Units per mL) and survival in the ic-MPT assay, we would argue that being able to distinguish between the amount of functional and non-functional antibodies required for clearance and protection in mice is critical for defining wP potency.

We employed a proteomic *B. pertussis* protein microarray containing 127 unique *B. pertussis* proteins cloned from Tahoma I strain to identify immunodominant antigens associated with wP vaccination in mice in an attempt to identify a serological signature that correlates with wP vaccine potency in mice. A caveat with the proteome microarray used in this study is that the array is biased toward outer membrane (OM) proteins, restricting the identification of other periplasmic and cytoplasmic immunogenic antigens. We still found that up to 71% of the wP immunodominant proteins identified in this study were also immunodominant (present in convalescent serum) in another study (Table 1). Of these, four pertussis antigens (BP3342, BP2992, BP0943, and BP0840) are important either for lung colonization in a baboon model or survival *in vitro*, indicating that they are not only immunogenic but contribute to persistence and could be important correlates of protection (27). One of the proteins we found in our study, BP1112 (BipA), is one of the most upregulated proteins in *B. pertussis* biofilms (41). De Gouw et al. reported that vaccination with recombinant BipA conferred protection against infection in mice, and anti-BipA antibodies efficiently opsonized *B. pertussis* (41). We also found that OmpA (BP0943) was universally reactive (100% seropositive) in the >100 serum samples tested. As it turns out, others have recently noted its reactivity and shown that anti-OmpA antibodies are immunogenic and even cross-reactive with *P. aeruginosa* (42).

Having established the validity of the proteome array, we then applied it to thermally stressed samples and identified differential reactivity profiles between $DTwP_4$ and $DTwP_{60,80,100}$. Surprisingly, we identified the loss of only two antigens, BP2497 (Zinc Protease) and BP3342 (PAL) when DTwP was subjected to higher temperatures. BP2497 has been implicated in several studies to be involved in the modulation of virulence genes. In another study, BP2497 was identified as a potential vaccine candidate, and immunization with recombinant BP2497 resulted in a small but significant reduction of bacterial load in the mice lungs on day 3. Therefore, it is reasonable that the loss of this protein in the thermally stressed vaccine was associated with higher bacterial lung colonization in our study. Pal was identified by Gregg et al. as a potential vaccine candidate because it was immunogenic, essential for persistence in the airway, and localized to the outer membrane. Hence, BP2497 and PAL are of great interest and could serve as markers of vaccine stability. On the other hand, we unexpectedly found five antigens of interest whose antibody reactivities increased with thermal stress. We speculate that heat treatment may result in the unmasking of other surface-associated proteins, such as the five identified in our studies, that would otherwise have remained masked in the absence of temperature stress. Nonetheless, the elevated antibody responses to these antigens are informative as possible biomarkers of DTwP vaccine instability or degradation.

We acknowledge several important limitations of this study that will need to be addressed in future endeavors. First, in the current study, forced degradation of wP was only conducted using thermal stress when, in fact, the World Health Organization recommends a range of "alteration" conditions (34). Second, we only used DTwP for the analysis when it is imperative that similar potency studies be conducted with bulk wP prior to being combined with additional components. Third, we relied on a single mouse strain for potency determinations when, in fact, numerous mouse strains are used by different global manufacturers (13). Finally, we envision expanding the *B. pertussis* proteome array to cover additional potency-determining antigens that will provide redundancy and breadth across different *B. pertussis* strains used in wP manufacturing.

In conclusion, this study provides relevant insights on the identification of antigenic biomarkers of wP vaccine stability and integrity, which directly impacts wP vaccine potency, as evidenced by relative differences in the protection provided by optimal and sub-optimal DTwP vaccines. The rational identification of these biomarkers could pave the way for the development of novel potency assays for pertussis vaccines to replace the ic-MPT and/or possibly serve at the patient level as surrogates of protection, particularly for immunocompromised patients. Our study also provides evidence that protection against wP infection is impacted by the production of good quality and functional antibody responses against *B. pertussis*.

## ACKNOWLEDGMENTS

We thank the Wadsworth Center's Veterinary Sciences for assistance in planning animal studies. We thank Dr. Kim Musser and members of the Wadsworth Center's Bacteriology Laboratory for assistance with *B. pertussis* cultures and Grace Freeman-Gallant for assisting with graphics. We thank Dr. Laura Viviani and members of the PSPT consortium for advice and feedback.

This work was supported by the Bill and Melinda Gates Foundation (awards INV-009301; AA-ID126). We gratefully acknowledge the Wadsworth Center's cell and tissue culture facility for providing bacterial media and ATGC core for whole-genome sequencing.

## AUTHOR AFFILIATIONS

[1]Division of Infectious Diseases, Wadsworth Center, New York State Department of Health, Albany, New York, USA
[2]Department of Pharmaceutical Chemistry, Vaccine Analytics and Formulation Center, University of Kansas, Lawrence, Kansas, USA
[3]Antigen Discovery, Inc., Irvine, California, USA

## AUTHOR ORCIDs

Yetunde Adewunmi http://orcid.org/0000-0001-6947-152X
Jennifer Doering http://orcid.org/0000-0001-8749-774X
Nicholas J. Mantis http://orcid.org/0000-0002-5083-8640

## FUNDING

| Funder | Grant(s) | Author(s) |
| --- | --- | --- |
| Bill and Melinda Gates Foundation | INV-009301 | David B. Volkin |
| Bill and Melinda Gates Foundation | AA-ID126 | Joseph J. Campo |

## AUTHOR CONTRIBUTIONS

Yetunde Adewunmi, Formal analysis, Investigation, Methodology, Writing – original draft, Writing – review and editing | Jozelyn V. Pablo, Formal analysis, Methodology | Andy A. Teng, Formal analysis, Investigation, Methodology | Vu Huynh, Formal analysis, Methodology | Kathryn Secrist, Investigation, Methodology.

## ADDITIONAL FILES

The following material is available online.

### Supplemental Material

**Supplemental figures (Spectrum03253-24-s0001.pdf).** Figures S1 to S3.
**Tables S1 to S9 (Spectrum03253-24-s0002.xlsx).** Excel file with proteome analysis.

## Open Peer Review

**PEER REVIEW HISTORY (review-history.pdf).** An accounting of the reviewer comments and feedback.

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
