## [Reviewer comments · Microbiology Spectrum]

Microbiology Spectrum

Antibody Profiles Elicited by Potent and Subpotent Whole Cell Pertussis Vaccines in Mice

Yetunde Adewunmi, Jennifer Doering, Prashant Kumar, Jozelyn Pablo, Andy Teng, Vu Huynh, Kathryn Secrist, David Volkin, Sangeeta Joshi, Joseph Campo, and Nicholas Mantis

Corresponding Author(s): Nicholas Mantis, Wadsworth Center, New York State Department of Health

Review Timeline:

Submission Date:	January 3, 2025
Editorial Decision:	January 24, 2025
Revision Received:	January 24, 2025
Accepted:	January 27, 2025

Editor: Artem Rogovsky

Reviewer(s): The reviewers have opted to remain anonymous.

Transaction Report:

DOI: <https://doi.org/10.1128/spectrum.03253-24>

Re: Spectrum03253-24 (Antibody Profiles Elicited by Potent and Subpotent Whole Cell Pertussis Vaccines in Mice)

Dear Dr. Nicholas J. Mantis:

Thank you for the privilege of reviewing your work. Below you will find my comments, instructions from the Spectrum editorial office, and the reviewer comments.

I am pleased to inform you that your manuscript has been editorially accepted for publication. However, there are a few additional questions in the submission form that need to be answered before the final decision. Once these are completed, please return your submission so that I can move your paper forward to acceptance.

Revision Guidelines

To submit your modified manuscript, log into the submission site at <https://spectrum.msubmit.net/cgi-bin/main.plex>. Go to Author Tasks and click the appropriate manuscript title to begin. The information you entered when you first submitted the paper will be displayed; update this as necessary.

Sincerely,
Artem Rogovsky
Editor
Microbiology Spectrum

Re: Spectrum03253-24R1 (Antibody Profiles Elicited by Potent and Subpotent Whole Cell Pertussis Vaccines in Mice)

Dear Dr. Nicholas J. Mantis:

Your manuscript has been accepted, and I am forwarding it to the ASM production staff for publication. Your paper will first be checked to make sure all elements meet the technical requirements. ASM staff will contact you if anything needs to be revised before copyediting and production can begin. Otherwise, you will be notified when your proofs are ready to be viewed.

Sincerely,
Artem Rogovsky
Editor
Microbiology Spectrum